# *Macromonas nakdongensis* sp. nov., Isolated from Freshwater and Characterization of Bacteriophage BK-30P—The First Phage That Infects Genus *Macromonas*

**DOI:** 10.3390/microorganisms11092237

**Published:** 2023-09-05

**Authors:** Kiwoon Baek, Ahyoung Choi

**Affiliations:** 1Department of Biological Sciences and Bioengineering, Inha University, Incheon 22212, Republic of Korea; backy8575@nnibr.re.kr; 2Nakdonggang National Institute of Biological Resources, 137 Donam 2-gil, Sangju 37242, Republic of Korea

**Keywords:** *Macromonas nakdongensis*, novel species, Macromonasphage, *Macromonas* bacteriophage, genome, freshwater

## Abstract

A Gram-stain-negative, non-motile, non-pigmented, rod-shaped bacterium was isolated from a freshwater sample of Nakdong River in South Korea and designated as strain BK-30^T^. An analysis of the 16S rRNA gene sequence of strain BK-30^T^ revealed its closest phylogenetic neighbors were members of the genus *Macromonas*. Specifically, the strain formed a robust clade with *Macromonas bipunctata* DSM 12705^T^, sharing 98.4% similarity in their 16S rRNA gene sequences. The average nucleotide identity value between strain BK-30^T^ and *M. bipunctata* DSM 12705^T^ was 79.8%, and the genome-to-genome distance averaged 21.3%, indicating the representation of a novel genomic species. Strain BK-30^T^ exhibited optimum growth at 30 °C and pH 7.0, in the absence of NaCl. The major respiratory isoprenoid quinone identified was ubiquinone-8 (Q-8). The principal fatty acids detected were C_16:1_ *ω*7*c* and/or C_16:1_ *ω*6*c* (49.6%), C_16:0_ (27.5%), and C_18:1_ *ω*7*c* and/or C_18:1_ ω6c (9.2%). The DNA G+C content of the strain was determined to be 67.3 mol%. Based on these data, we propose a novel species within the genus *Macromonas*, named *Macromonas nakdongensis* sp. nov., to accommodate the bacterial isolate. Strain BK-30^T^ is designated as the type strain (=KCTC 52161^T^ = JCM 31376^T^ = FBCC-B1). Additionally, we present the isolation and complete genome sequence of a lytic phage infecting strain BK-30^T^, named BK-30P. This bacteriophage is the first reported to infect *Macromonas*, leading us to propose the name “Macromonasphage”.

## 1. Introduction

The genus *Macromonas*, belonging to the family *Comamonadaceae* [1], is classified within the class *Betaproteobacteria*. The genus *Macromonas* currently consists of two species: *Macromonas mobilis* and *Macromonas bipunctata*. The first proposal of the genus *Macromonas* included the description of *Macromonas mobilis* [2,3], although this species is not available in culture [4]. Afterward, *Macromonas bipunctata*, another species of the genus *Macromonas*, was described by Dubinina and Grabovich [5] with a strain isolated from the precipitates of sewage aeration tanks. Due to the absence of available cultures and 16S rRNA gene sequences for *M. mobilis*, only *Macromonas bipunctata* has been designated as the type species of the genus for taxonomic classification [6,7]. The type strain *Macromonas bipunctata* IAM 14880^T^ exhibits characteristics such as being Gram-stain-negative, non-pigmented, oxidase-positive, catalase-negative, motile due to polar flagella, and aerobic. The cells of *Macromonas bipunctata* are irregular and rod-shaped. The DNA G+C content of *M. bipunctata* is 63.8 mol%. Additionally, *M. bipunctata* predominantly contains ubiquinone-8 (Q-8) as the major respiratory quinone.

Bacteriophages that can lyse specific bacterial hosts play a crucial role in controlling bacterial diversity and abundance, influencing biogeochemical cycles. Therefore, studies into diverse viruses are important with various ecological relevance [8]. To the best of our knowledge, no lytic phage infecting *Macromonas* strains has been reported, possibly due to the lack of suitable bacterial hosts [9].

In this study, we describe a bacterial strain BK-30^T^, which was isolated from freshwater, as a representative of a novel species within the genus *Macromonas*, based on taxonomic data obtained using polyphasic approaches. An analysis of the bacterial metadatabase BacDive (http://bacdive.dsmz.de/, accessed on 28 August 2023) revealed the global distribution of the genus *Macromonas*, spanning diverse geographical locations [10]. Remarkably, a substantial proportion, exceeding 81.4%, of these occurrences were observed in aquatic samples. Despite its widespread prevalence, the limited identification of species within the genus *Macromonas* poses a challenge to its comprehensive characterization. Our study introduces a significant advancement by reporting the discovery of a novel species within the genus *Macromonas*. Notably, this discovery is the first of its kind, originating from a freshwater environment, thereby exemplifying an important aquatic sample type. This finding contributes to a more refined understanding of the genus and its ecological adaptations. Furthermore, we present the isolation and genomic characterization of phage BK-30P, a novel podovirus found in freshwater that infects *Macromonas* sp. strain BK-30^T^. The genome features and annotation of phage BK-30P are discussed. These findings provide insights into the host strain’s capacity to interact with the environmental diversity of *Macromonas*-associated phages.

## 2. Materials and Methods

### 2.1. Isolation of Bacterial Strain

Strain BK-30^T^ was isolated from a freshwater sample collected from Nakdong River in Korea (36°26′33″ N, 128°15′24″ E). The isolation was performed using a standard dilution plating method on R2A agar (BD Difco, Franklin Lakes, NJ, USA) followed by incubation at 20 °C for one month. After determining the optimal growth temperature, working cultures of strain BK-30^T^ were maintained on R2A agar at 30 °C. To preserve the strain, glycerol suspensions (20% in distilled water, *v*/*v*) were prepared, and the cultures were stored at −80 °C. For phenotypic comparison, the reference strain *M. bipunctata* DSM 12705^T^ obtained from DSMZ (Deutsche Sammlung von Mikroorganismen und Zellkulturen, Braunschweig, Germany) was used. It was also maintained on R2A agar at 30 °C.

### 2.2. 16S rRNA Gene Phylogeny

For 16S rRNA gene sequence analysis, the genomic DNA of strain BK-30^T^ was extracted using the DNeasy Blood and Tissue kit (Qiagen, Hilden, Germany), following the manufacturer’s instructions. The 16S rRNA gene was amplified using the primers 27F and 1492R [11] and then sequenced using four primers, 27F, 518F, 907R, and 1492R, using Sanger sequencing. The obtained nearly complete 16S rRNA gene sequence was compared to sequences in the GenBank database using BLAST [12], using the EzBioCloud database (www.ezbiocloud.net/eztaxon, accessed on 28 August 2023) [13] for taxonomic identification. The sequences of the 16S rRNA gene aligned in the silva aligner ver1.2.1.11 (http://www.arb-silva.de/aligner, accessed on 28 August 2023) [14] were opened in MEGA 7.0 [15], manually checked for alignment, and used for phylogenetic analyses. Phylogenetic trees were generated based on the neighbor-joining method [16] using the algorithms of Jukes–Cantor distance [17], maximum parsimony [18], and maximum likelihood [19]. The robustness of the neighbor-joining, maximum likelihood, and maximum parsimony trees was confirmed by bootstrap analyses based on 1000 random resamplings.

### 2.3. Polyphasic Taxonomy

Phenotypic characteristics of strain BK-30^T^ and the reference strain *M. bipunctata* DSM 12705^T^ were determined using bacterial cultures grown under the same conditions. The temperature range and optimal temperature for growth were determined by culturing in R2A broth (MB cell) at 5–45 °C (5 °C, 10 °C, 15 °C, 20 °C, 25 °C, 30 °C, 35 °C, 40 °C, and 45 °C) for 10 days. The pH range and optimum pH for growth were determined at pH 5.0–10.0 (at intervals of 0.5 pH unit) in R2A broth. The pH was adjusted using the following buffers: MES (pH 5.0–6.0), MOPS (pH 6.5–7.0), HEPES (pH 7.5–8.0), Tris (pH 8.5–9.0), and CHES (pH 9.5–10.0), each at a final concentration of 50 mM for up to 10 days. Salt requirement and tolerance were determined on R2A broth supplemented with 0–3.0% (at intervals of 0.5%), 4.0, 5.0, 7.5, 10.0, and 15.0% (*w*/*v*) NaCl. For testing growth on reduced sulfur and elemental sulfur, strain BK-30^T^ was inoculated into R2A broth, where the pH of which was adjusted to 7.0, and supplemented with elemental sulfur (1%), sulfide (1%), thiosulfate (1%), or sulfite (1%). The growth of each culture was monitored by measuring turbidity via spectrophotometry (Optizen 2120UV; Mechasis, Ankara, Türkiye) at 12 h intervals over a 5-day period.

Gram staining was performed using a Gram staining kit (Sigma), and growth under anaerobic conditions was examined using the MGC anaerobic system with Anaero-PACK (Mitsubishi Gas Chemical, Tokyo, Japan). Gliding motility was investigated using phase-contrast microscopy and the hanging drop method [20]. Cell morphology, cell size, and the presence of flagella were examined using transmission electron microscopy (TEM) (CM200; Phillips) by staining the bacterial cells with 2.0% uranyl acetate on a carbon-coated copper grid. The production of H_2_S was determined using triple sugar iron agar (BD Difco), and hydrolysis tests for Tweens 20, 40, 60, and 80 were performed on R2A supplemented with each component. The hydrolysis of casein, tyrosine, CM-cellulose, and starch was assessed based on the formation of clear zones around colonies after applying suitable staining solutions. DNase test agar (BD Difco) was used to evaluate DNA degradation. The degradation of hypoxanthine and xanthine was tested using R2A supplemented with each component, following the protocol of Gordon et al. [21]. Other biochemical tests and carbon source utilization tests were conducted using API 20NE, API ZYM (BioMérieux, Marcy-l’Étoile, France), and GN2 microplates (Biolog, Hayward, CA, USA) according to the manufacturer’s instructions, with inocula prepared in phosphate-buffered saline (pH 7.4).

Fatty acids of strain BK-30^T^ and *M. bipunctata* DSM 12705^T^ were extracted from cultures grown on R2A at 30 °C in the late exponential phase (3 days). An analysis was performed using the Sherlock Microbial Identification System version 6.1 (MIDI) with the TSBA database. The respiratory isoprenoid quinones were purified by TLC according to the protocol described by Minnikin et al. [22] and analyzed using HPLC [23].

### 2.4. Genome Sequencing, Assembly, and Annotation

For genome sequencing, the genomic DNA of strain BK-30^T^ was extracted using the DNeasy Blood and Tissue kit (Qiagen), following the manufacturer’s instructions. The genome sequencing of strain BK-30^T^ was performed using the Illumina MiSeq platform (Macrogen Inc., Seoul, Republic of Korea). The de novo assembly of the genome using CLC Workbench^®^ v9.0 (CLC Bio-Qiagen) resulted in 182 contigs, with an average genome coverage of 110.0×. The genome sequence of strain *M. bipunctata* DSM 12705^T^ was obtained using the same procedures. The assembled genome sequences satisfied the minimal standards for being used for taxonomic purposes [24]. The assemblies were then submitted to rapid annotation using the Subsystem Technology (RAST) server [25].

The genomic relatedness was estimated based on an average nucleotide identity (ANI) using the OrthoANI algorithm in the EzBioCloud service (http://www.ezbiocloud.net/tools/ani/, accessed on 20 July 2023) [26] and digital DNA-DNA hybridization (dDDH) values using the genome-to-genome distance calculator (GGDC) version 2.1 (http://ggdc.dsmz.de/distcalc2.php, accessed on 28 August 2023) [27]. To infer a genome-based phylogenetic tree, the universal core gene set was extracted using the up-to-date bacterial core gene set (UBCG) pipeline [28] and subjected to FastTree [29] to reconstruct a phylogenetic tree based on amino acid alignment.

### 2.5. Phage Isolation and Genomic DNA Preparation

Macromonasphage BK-30P was isolated from a surface freshwater sample collected from Nakdong River during the spring of 2017. The isolation process involved the standard plaque assay after enriching the sample with the host bacterium. To initiate enrichment, the river water sample was first filtered through 0.22 μm pore size membrane filters (Durapore, Millipore, Burlington, MA, USA) to remove bacterial particles. Into the filtrate (400 mL), the host culture (strain BK-30^T^; 30 mL) and 5 × R2A broth (100 mL; autoclaved) were added, followed by incubation at 30 °C for 5 days. Subsamples of the enrichment culture were treated with chloroform. The presence of phages in the enrichment culture was then checked using the double agar overlay plaque assay with 0.6% low-melt top agar, which contained an actively growing host culture (10 mL) and the chloroform-treated enrichment culture (3 mL). After 3 days of incubation, plaques with diameters ranging from 2 to 3 mm were visible on the plates. A clear plaque was selected, followed by pure isolation through three times of repeated plaque assays and single plaque picking. This isolated phage was designated as BK-30P.

The purification of phage DNA was carried out following a modified version of the procedure described in *Molecular Cloning: A Laboratory Manual* [30]. About 2 L of phage lysates was prepared for the DNA purification process. To these lysates, DNase I and RNase A were added at a final concentration of 1 μg per mL. Subsequently, 116.9 g NaCl was dissolved in the lysates and the mixture was cooled to 4 °C. After an incubation period of approximately 1 h, the combined lysates were subjected to centrifugation at 10,000× *g* for 30 min at 4 °C to eliminate debris. The phage particles present in the resulting supernatant were then precipitated using 10% (*w*/*v*) PEG 8000 (Sigma-Aldrich, Burlington, MA, USA). Following overnight incubation at 4 °C, the mixture was subjected to centrifugation at 10,000× *g* for 25 min at 4 °C, and the resulting pellet was delicately resuspended in 2 mL of SM buffer (composed of 50 mM Tris–HCl, pH 7.5; 100 mM NaCl; 10 mM MgSO_4_·7H_2_O; 0.01% gelatin). The removal of PEG was achieved by treating the sample with an equal volume of chloroform. Subsequently, the phages were subjected to ultracentrifugation at 246,000× *g* for 2 h at 4 °C using an L-90 K ultracentrifuge (Beckman, Brea, CA, USA) equipped with an SW 55 Ti rotor. The resulting pellet of phage particles was resuspended using 100 μL of SM buffer and incubated overnight at 4 °C. The purified phages were then stored in the dark at 4 °C. The genomic DNA of the BK-30P phage was isolated using a silica-based spin column (DNeasy Blood and Tissue Kit, Qiagen, Shanghai, China) in accordance with the manufacturer’s provided instructions.

### 2.6. Morphological and Genomic Characterization of Macromonasphage BK-30P

For morphological characterization, purified phage particles were adsorbed onto 200-mesh carbon-coated copper grids (Electron Microscopy Sciences, Hatfield, PA, USA) and negatively stained with 2% uranyl acetate for 10 s. The stained phage particles were examined using a transmission electron microscope (CM200, Phillips, Eindhoven, The Netherlands) to analyze their morphology.

To extract the genomic DNA of BK-30P, it was sequenced at Macrogen Inc. using an Illumina Miseq system with 2 × 300 bp paired-end reads. The obtained raw data were assembled using SPAdes version 3.1.1 [31]. The Illumina platform provided 82-fold coverage of the genome. The genome was assembled into one contig through PCR-based gap closing.

Gene prediction in the genome was performed using a combination of two gene calling methods: the RAST server [25] and Genemark.hmm 3.25 [32]. The assignment of protein function to ORFs was performed manually using BLASTp against the NCBI nonredundant database and RPS-BLAST or HMMER search against the COG database [33], Pfam database [34], and TIGRFam database [35]. Furthermore, transmembrane helices were predicted using TMHMM [36], while the anticipation of signal peptides was conducted through SignalP [37].

## 3. Results

### 3.1. 16S rRNA Gene Phylogeny

The nearly complete 16S rRNA gene sequence of strain BK-30^T^ (KU360711; GenBank accession number) was obtained and compared to other sequences by BLAST in GenBank and was also analyzed using the EzBioCloud database. Strain BK-30^T^ was found to be most closely related to *Macromonas bipunctata* IAM 14880^T^ (98.4% sequence similarity). Phylogenetic analysis based on the neighbor-joining method revealed that strain BK-30^T^ and *M. bipunctata* IAM 14880^T^ formed a robust clade, indicating their close relationship within the genus *Macromonas* (Figure 1).

### 3.2. Polyphasic Taxonomic Characterization

The cells of strain BK-30^T^ were Gram-stain-negative and rod-shaped (0.5–0.7 μm × 1.0–1.6 μm) (Appendix A). Growth occurs at 5–35 °C (optimum 30 °C), pH 6.0–8.0 (optimum pH 7.0), and 0–0.5% NaCl (optimum 0%). Strain BK-30^T^ grew chemolithotrophically on sulfide, sulfite, and elemental sulfur, but growth was not observed with thiosulfate. Hydrolysis occurred for tyrosine, hypoxanthine, xanthine, Tween 20, and Tween 40, but not for starch, DNA, CM-cellulose, casein, or Tween 80. Aesculin hydrolysis and gelatin liquefaction are positive, but nitrate reduction, indole production, glucose fermentation, arginine dihydrolase, urease, and PNPG (*β*-galactosidase) are negative (In API 20NE). With regard to API ZYM, it is positive for alkaline phosphatase, esterase (C4), lipase (C8), leucine arylamidase, valine arylamidase, and naphthol-AS-BI-phosphohydrolase, but negative for esterase lipase (C14), cysteine arylamidase, trypsin, *α*-chymotrypsin, acid phosphatase, *α*-galactosidase, *β*-galactosidase, *β*-glucuronidase, α-glucosidase, *β*-glucosidase, *N*-acetyl-*β*-glucosaminidase, *α*-mannosidase, and *α*-fucosidase. It is also positive for dextrin, Tween 40, D-fructose, *α*-D-glucose, L-rhamnose, sucrose, pyruvic acid methyl ester, D-galacturonic acid, *α*-hydroxybutyric acid, *α*-keto butyric acid, *α*-ketoglutaric acid, DL-lactic acid, propionic acid, succinic acid, bromosuccinic acid, L-alanine, L-glutamic acid, L-phenylalanine, L-proline, L-serine, L-threonine, *γ*-aminobutyric acid, and glycerol using the Biolog GN2 carbon source oxidation test. A comparative phenotypic analysis revealed several differences between strain BK-30^T^ and *M. bipunctata* DSM 12705^T^ in terms of cell shape, enzyme activities, carbon source oxidation, and macromolecule hydrolysis (Table 1).

The major fatty acids (>10%) identified in strain BK-30^T^ were summed feature 3 (C_16:1_ *ω*6*c* and/or C_16:1_ *ω*7*c*) (49.6%) and C_16:0_ (27.5%) (Table 2), which were similar to the fatty acid composition of *M*. *bipunctata* DSM 12705^T^. The predominant respiratory quinone detected in strain BK-30^T^ was indeed ubiquinone-8 (Q-8).

### 3.3. Genome Characterization

The draft genome sequence of strain BK-30^T^ (NWMV00000000; GenBank accession number) was determined to be 3,064,983 bp in length. The genomic DNA G+C content of strain BK-30^T^ was found to be 67.3%, which is slightly higher than that of *Macromonas bipunctata* DSM 12705^T^ (63.8 mol%). The genome annotation predicted a total of 2893 coding sequences (CDSs), 53 tRNA genes, and 3 rRNA genes (Appendix A). A RAST analysis (https://rast.nmpdr.org, accessed on 28 August 2023) was performed to predict the functional gene content of the strain SJOD-M-6^T^ genome. The analysis revealed the presence of genes involved in various biological processes, including amino acids and derivatives (340 genes), carbohydrates (271 genes), cofactors, vitamins, prosthetic groups, and pigments (235 genes), as well as protein metabolism (235 genes) (Appendix A). We also confirmed the presence of various genes related to sulfur oxidation (*Sox*) in the genome of strain BK-30^T^. The sulfur oxidation system, which includes sulfite dehydrogenase, sulfide dehydrogenase, and sulfite oxidase, is involved in the oxidation of reduced sulfur compounds. These genes are part of the *Sox* cluster, specifically the *soxABXYZDF* cluster, which encodes the multienzyme *Sox* complex responsible for the complete oxidation of sulfur compounds.

Whole-genome sequence comparisons were performed to determine the genomic relatedness between strain BK-30^T^ and *M. bipunctata* IAM 14880^T^. The genomic relatedness analysis using ANI values and dDDH values supports the conclusion that strain BK-30^T^ represents a distinct species within the genus *Macromonas* and is different from *M. bipunctata* DSM 12705^T^. The ANI value of 79.9% and the dDDH value of 21.9% are below the commonly accepted thresholds for species demarcation [24,27,38]. Therefore, based on these results, strain BK-30^T^ can be considered a novel species separate from *M. bipunctata*. Furthermore, the phylogenomic tree showed that the genome of strain BK-30^T^ formed a clade with the genome of *M. bipunctata*, confirming again that strain BK-30^T^ represents a novel species of the genus *Macromonas* (Figure 2). The overall genomic characteristics, particularly the ANI and dDDH values, suggest that strain BK-30^T^ represents a novel species of the genus *Macromonas*.

### 3.4. Macromonasphage Analysis

Morphological characterization using TEM showed that phage BK-30P is a podovirus, a group of double-stranded DNA (dsDNA) bacteriophages with icosahedral heads and short tails. The dimensions of the phage were determined to be approximately 64 nm for the head diameter and 5–7 nm for the tail length (Figure 3).

The genome of BK-30P (OR420333; GenBank accession number) was sequenced and assembled resulting in a 43,064 bp dsDNA molecule with a G+C content of 58.6%. No tRNA-encoding genes were identified in the genome (Appendix A). A total of 60 open reading frames (ORFs) were predicted in the phage genome. Among these, 21 genes were assigned putative functions, while the remaining genes were annotated as coding for hypothetical proteins. Four proteins with transmembrane helices were identified, but signal peptides were not detected in any protein (Appendix A). The functions of the identified genes were associated with various processes such as DNA packaging, replication, and metabolism, as well as head and tail assembly (Appendix A). Additionally, a phylogenetic tree constructed using the *TerL* gene demonstrated the close relationship of phage BK-30P with some podoviruses previously classified as the family *Podoviridae* (Appendix A).

## 4. Discussion

The taxonomic characterization of strain BK-30^T^ is based on a polyphasic approach, including 16S rRNA gene sequence analysis, physiological tests, fatty acid composition, and genomic analysis. The 16S rRNA gene analysis indicates that strain BK-30^T^ is most closely related to *Macromonas bipunctata* DSM 12705^T^, sharing 98.4% sequence similarity. However, other phenotypic and genomic characteristics differentiate BK-30^T^ from *M. bipunctata* DSM 12705^T^. These differences include growth temperature, pH range, carbon source utilization, enzyme activities, and fatty acid composition. Furthermore, genomic comparison using ANI and dDDH values confirms that BK-30^T^ represents a novel genomic species within the genus *Macromonas*. The ANI value of 79.9% and the dDDH value of 21.9% fall below the established thresholds for species delineation. This comprehensive taxonomic analysis supports the proposal of *Macromonas nakdongensis* sp. nov. as being a distinct species within the genus.

The discovery of a novel *Macromonas* species, particularly from a freshwater environment like Nakdong River, is of ecological significance. The genus *Macromonas* has been found in various aquatic habitats, making this finding especially relevant to understanding its diversity and ecological adaptations. By expanding the knowledge of the *Macromonas* species, this study contributes to a more nuanced understanding of microbial communities in freshwater ecosystems and their potential ecological roles.

The manuscript goes beyond bacterial taxonomic characterization by introducing the isolation and characterization of a novel lytic phage, BK-30P, which infects *Macromonas* sp. strain BK-30^T^. This represents a notable advancement, as no *Macromonas*-specific phage had previously been reported. The detailed morphological and genomic characterization of BK-30P sheds light on the diversity of phages that interact with the genus *Macromonas*. The identification of BK-30P as a podovirus adds to the understanding of phage diversity and the potential impacts of phages on bacterial communities in freshwater environments.

However, we do recognize the limitations of this study. The slow growth and small colonies of strain BK-30^T^ presented challenges for a comprehensive genome analysis. Despite these constraints, our discoveries hold significant taxonomic and ecological implications.

## 5. Conclusions

In conclusion, our study introduces a novel species within the genus *Macromonas, Macromonas nakdongensis*, isolated from freshwater in Nakdong River. This discovery contributes to our understanding of the taxonomic diversity and ecological relevance of the *Macromonas* genus. Furthermore, the isolation of bacteriophage BK-30P highlights the complex interactions between phages and their hosts in aquatic environments. As we continue to unravel the intricacies of microbial communities in freshwater habitats, our findings lay the foundation for future investigations into the ecological roles and dynamics of *Macromonas* and its associated phages.

## 6. Protologue

Description of *Macromonas nakdongensis* sp. nov.

*Macromonas nakdongensis* (nak.dong.en′sis. N.L. masc. adj. *nakdongensis* from Nakdong River, where the type strain was isolated).

It is Gram-stain-negative, oxidase, catalase-positive, non-motile, non-gliding, and strictly aerobic. Cells are rod-shaped (0.5–0.7 μm × 1.0–1.6 μm). Colonies are 0.7–1.0 mm in diameter, circular, convex, smooth, shiny, transparent with an entire margin, and white-colored after incubation for 3 days at 30 °C on R2A agar. Growth occurs at 5–35 °C (optimum 30 °C), pH 5.0–8.0 (optimum pH 7.0), and in the presence of 0–1.0% NaCl (optimum 0%). Grows chemolithotrophically on sulfide, sulfite, and elemental sulfur, but not on thiosulfate. Tyrosine, hypoxanthine, xanthine, Tween 20, and Tween 40 are hydrolyzed, but starch, DNA, CM-cellulose, casein, and Tween 80 are not hydrolyzed. H2S is not produced. It is positive for aesculin hydrolysis and gelatinase liquefaction, but negative for nitrate reduction, indole production, glucose fermentation, arginine dihydrolase, urease, and PNPG (*β*-galactosidase) (in API 20NE). With regard to API ZYM, it is positive for alkaline phosphatase, esterase (C4), lipase (C8), leucine arylamidase, valine arylamidase, and naphthol-AS-BI-phosphohydrolase, but negative for esterase lipase (C14), cystine arylamidase, trypsin, α-chymotrypsin, *α*-galactosidase, *β*-galactosidase, *α*-glucosidase, *β*-glucosidase, N-acetyl-*β*-glucosaminidase, *α*-mannosidase, and *α*-fucosidase. It is positive for dextrin, Tween 40, D-fructose, *α*-D-glucose, L-rhamnose, sucrose, pyruvic acid methyl ester, D-galacturonic acid, *α*-hydroxybutyric acid, *α-keto butyric* acid, α-ketoglutaric acid, DL-lactic acid, propionic acid, succinic acid, bromosuccinic acid, L-alanine, L-glutamic acid, L-phenylalanine, L-proline, L-serine, L-threonine, *γ*-aminobutyric acid, and glycerol using the Biolog GN2 source oxidation test. The isoprenoid quinone detected is Q-8. The major cellular fatty acids are summed feature 3 (C_16:1_ *ω*6*c* and/or C_16:1_ *ω*7*c*) (49.6%) and C_16:0_ (27.5%).

The type strain of *Macromonas nakdongensis* is BK-30^T^ (=KCTC 52161^T^ = JCM 31376^T^ = FBCC-B1), which was isolated from a freshwater sample of Nakdong River in Korea. The G+C content of the type strain DNA is 67.3 mol%, and its genome size is 3.1 Mb. The 16S rRNA gene sequence is deposited in GenBank/EMBL/DDBJ under the accession number KU360711, and the draft genome sequence is available under the accession number NWMV00000000.

## Figures and Tables

**Figure 1 microorganisms-11-02237-f001:**
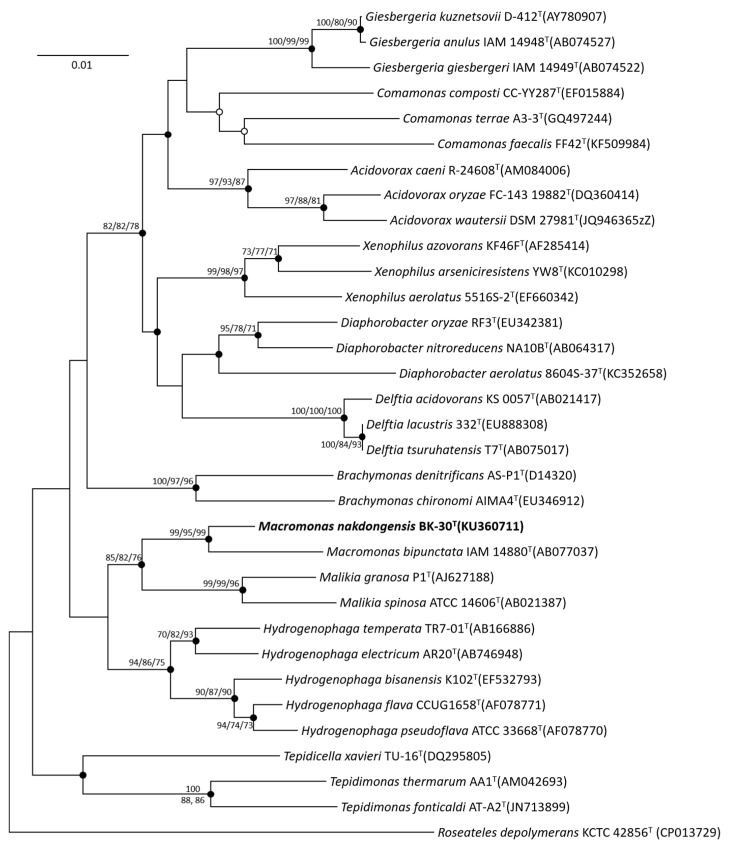
Neighbor-joining phylogenetic tree based on 16S rRNA gene sequences, showing the relationship between strain BK-30^T^ and other representative type species of the family *Comamonadaceae*. Bootstrap values (expressed as percentages of 1000 replications) over 70% are shown at nodes for neighbor-joining, maximum likelihood and maximum parsimony methods, respectively. Filled circles indicate that the corresponding nodes were recovered by all treeing methods. An open circle indicates that the corresponding node was recovered by the neighbor-joining and maximum likelihood methods. Bar, 0.01 substitution per nucleotide position.

**Figure 2 microorganisms-11-02237-f002:**
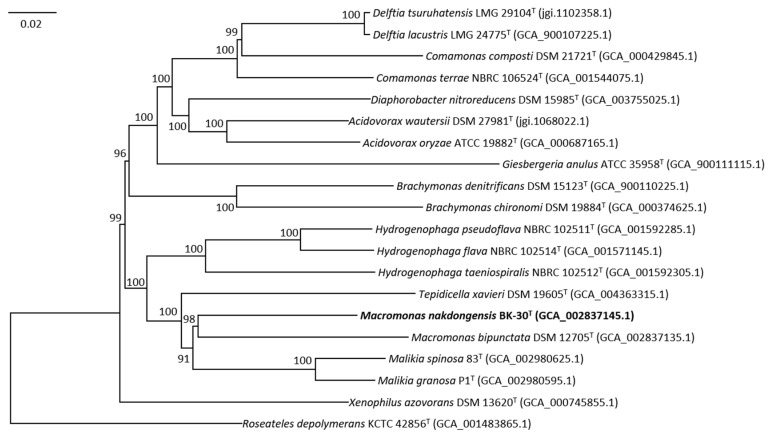
Phylogenomic tree based on concatenated multiple alignment of 84 genes showing the relationship between BK-30^T^ and members of the family *Comamonadaceae*. Bootstrap values based on 100 replicates are shown at nodes. The bar represents 0.02 substitutions per nucleotide position.

**Figure 3 microorganisms-11-02237-f003:**
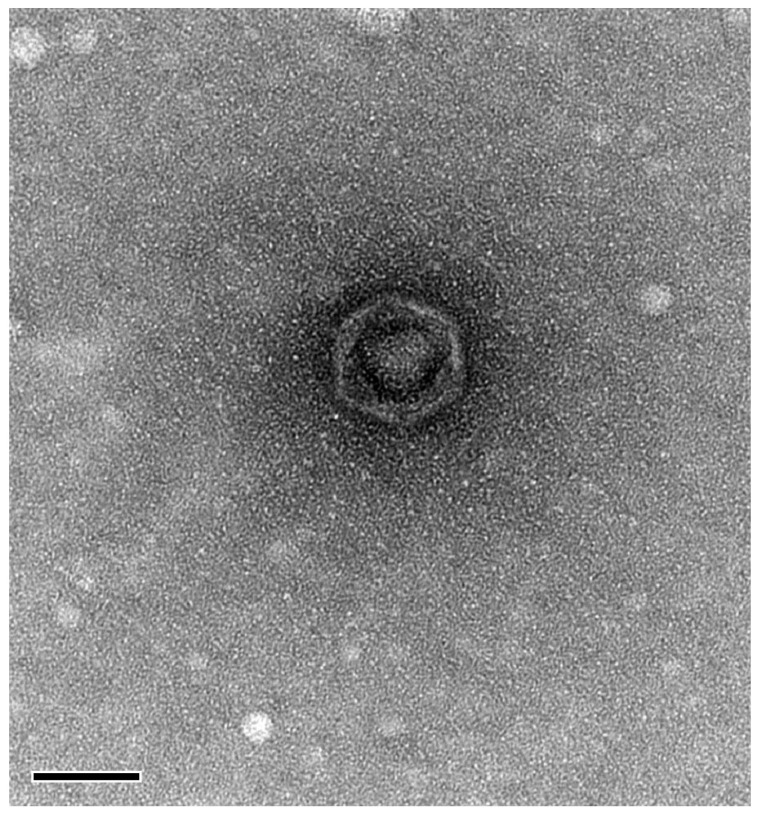
Transmission electron micrograph of Macromonasphage BK-30P. The scale bar is 50 nm.

**Table 1 microorganisms-11-02237-t001:** Major characteristics that differentiate between strain BK-30^T^ and *Macromonas bipunctata* DSM 12705^T^.

Characteristic	1	2
Cell shape	Rod	Irregular rod
Temperature range for growth (°C)	5–35	10–35
Flagella	-	Polar tuft *
API 20EN:		
Gelatinase	+	-
Enzyme activities (API ZYME):		
Valine	+	-
Acid phosphatase and *β*-glucuronidase	-	+
Utilization of carbon sources (Biolog GN2):		
dextrin, Tween 40, D-fructose, *α*-D-glucose, L-rhamnose, sucrose, pyruvic acid methyl ester, D-galacturonic acid, α-hydroxybutyric acid, *α*-keto butyric acid, α-ketoglutaric acid, DL-lactic acid, propionic acid, succinic acid, bromosuccinic acid, L-alanine, L-glutamic acid, L-phenylalanine, L-proline, L-serine, L-threonine, *γ*-aminobutyric acid, glycerol	+	-
Hydrolysis of the following:		
tyrosine, hypoxanthine, xanthine, Tween 20, Tween 40	+	-
DNA G+C content (%)	67.3	63.8

Strain: 1, BK-30^T^; 2, *M. bipunctata* DSM 12705^T^. All data were obtained in this study unless otherwise indicated. +, positive; -, negative. * Data were taken from Spring et al. [3].

**Table 2 microorganisms-11-02237-t002:** Fatty acid profiles of strains BK-30^T^ and *M. bipunctata* DSM12705^T^.

Fatty Acid	1	2
Straight-chain		
C_12:0_	2.5	-
C_14:0_	tr	5.1
C_15:0_	1.2	-
C_16:0_	27.5	22.4
C_17:0_	1.6	-
C_18:0_	1.0	tr
Non-straight-chain		
C_15:1_ *ω*6*c*	1.0	-
C_17:1_ *ω*8*c*	1.8	-
Hydroxy		
C_13:0_ 2OH	tr	1.5
Branched-chain		
anteiso-C_17:0_	-	2.1
Summed features *		
3 (C_16:1_ *ω*6*c* and/or C_16:1_ *ω*7*c*)	49.6	56.1
7 (C_19:0_ *ω*6*c* and/or C_19:0_ cyclo *ω*7*c*)	2.4	-
8 (C_18:1_ *ω*6*c* and/or C_18:1_ *ω*7*c*)	9.2	11.3

Strains: 1, BK-30^T^; 2, *M. bipunctata* DSM12705^T^. Data were taken from this study using cells grown on R2A agar at 30 °C until the late exponential phase. Values are percentages of the total fatty acids. Fatty acids that represented <1.0% in all strains are not shown. tr, trace (<1.0%); -, not detected. * Summed features represent groups of two or three fatty acids that could not be separated via GLC using the MIDI system.

## Data Availability

The GenBank/EMBL/DDBJ accession number for the 16S rRNA gene sequence of strain *Macromonas nakdongensis* BK-30^T^ is KU360711. The draft genome sequence of strain BK-30^T^ and *M. bipunctata* DSM 12705^T^ are NWMV00000000 and NWBQ00000000, respectively. Additionally, the complete genome sequence of bacteriophage BK-30P is OR420333.

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
