# Peer review of "Macromonas nakdongensis sp. nov., Isolated from Freshwater and Characterization of Bacteriophage BK-30P—The First Phage That Infects Genus Macromonas"

_microorganisms, 2023, doi:10.3390/microorganisms11092237_

Round 1
Reviewer 1 Report
Introduction
The introduction is too generic. Most of the arguments were centered on the description of their Macromonas which I believe any average reader in the discipline should know already. Arguments should mainly be on the implications of the presence of this genus in the freshwater system. Also, nothing was said about the freshwater in which the bacterial strain BK-30T was isolated. This is very important. Then why this study? You mentioned you are proposing to present or you presented a new isolation and genomic characterization of phage BK-30P, a novel podovirus found in freshwater that infects Macromonas sp. strain BK-30T. kindly tell us more about how you came about the novel discovery, and what implications it will have on the scientific community. Overall, the introduction is too shallow and I would suggest you argue more about your research questions. Further, most of those descriptions of your isolate if you must add them to your write ups should be moved to the materials and methods section.
Materials and methods
This section was clear and well written.
Results and Discussion
This was fairly well written but can be improved upon. The presentation of the results should be more detailed with specific findings outlined. I am not comfortable with lumping of results and discussion together, it is confusing most times, so separate them. Further, it appears the authors did not consult more literature to compare with their findings. I would like to see arguments on the novel findings of this study and how this can enhance our science in this sub-discipline.
Conclusion
The conclusion was okay, but I would like it to be sharp and catchy, and it should dwell more on the novel findings, and very brief outlines on other aspect of the paper if you like.
Kindly see attached document for other comments.

The grammar is okay but a little edit is required when submitting the revised version of the manuscript.
Author Response
# Reviewer 1
Introduction
The introduction is too generic. Most of the arguments were centered on the description of their Macromonas which I believe any average reader in the discipline should know already. Arguments should mainly be on the implications of the presence of this genus in the freshwater system. Also, nothing was said about the freshwater in which the bacterial strain BK-30T was isolated. This is very important. Then why this study? You mentioned you are proposing to present or you presented a new isolation and genomic characterization of phage BK-30P, a novel podovirus found in freshwater that infects Macromonas sp. strain BK-30T. kindly tell us more about how you came about the novel discovery, and what implications it will have on the scientific community. Overall, the introduction is too shallow and I would suggest you argue more about your research questions. Further, most of those descriptions of your isolate if you must add them to your write ups should be moved to the materials and methods section.
→ (Response) Thank you for your insightful comments. We have carefully considered your feedback and have addressed the concerns by highlighting the significance of our discovery of a new Macromonas species in a freshwater environment
Materials and methods
This section was clear and well-written.
Results and Discussion
This was fairly well written but can be improved upon. The presentation of the results should be more detailed with specific findings outlined. I am not comfortable with lumping of results and discussion together, it is confusing most times, so separate them. Further, it appears the authors did not consult more literature to compare with their findings. I would like to see arguments on the novel findings of this study and how this can enhance our science in this sub-discipline.
→ (Response) We sincerely appreciate your valuable feedback. In light of your suggestions, we have taken steps to improve the manuscript. The Results and Discussion sections have been separated to enhance clarity and avoid confusion. Moreover, we have extended our literature review to provide a more comprehensive comparison of our findings with existing research. We have also emphasized the novel aspects of our study's results and their potential contributions to the advancement of our sub-discipline. Your input has been invaluable in enhancing the quality of our work.
Conclusion
The conclusion was okay, but I would like it to be sharp and catchy, and it should dwell more on the novel findings, and very brief outlines on other aspect of the paper if you like.
→ (Response) Some corrections have been made based on your comments.
- - - - - - - - - - - - - - - - - - - - - - - - - - - - - - - - - - - - - - - - - - - - - - - - - - - - - - - - - - - - - - - - - - -
Kindly see attached document for other comments.
- Introduction:
This statement is not clear: Macromonas bipunctata has been serving as the type species of the genus for taxonomic purposes [6, 7].
→ (Response) We have amended the following sentence to make it clearer.
“Due to the absence of available cultures and 16S rRNA gene sequences for M. mobilis, only Macromonas bipunctata has been designated as the type species of the genus for taxonomic classification [6, 7].”
- Materials and Methods
2.3. Polyphasic Taxonomy
add units, please.: 5, 10, 15, 20, 25, 30, 35, 40, and 45 °C
→ (Response) We added units.
What suggest this pH range to you? Why not a lesser range or wider range of pH?: The pH range and optimum pH for growth were determined at pH 5.0–10.0 (at intervals of 0.5 pH unit) in R2A broth.
→ (Response) Since the pH range and optimum pH of strain BK-30 were determined with the OD value, R2A broth was used. In addition, the reason why only the range of 5.0 to 10.0 was measured is that the experiment was conducted under the same conditions as the existing type strain, Macromonas bipunctata.
- Conclusion
I am wondering how this your study speaks to diversity. I think you should be silent about that and dwell more on the novel Macromonas you identified. Diversity is a different thing entirely. I didn't see analysis for diversity in the entire manuscript.
→ (Response) Thank you for your valuable comments. In response to your suggestion, we have restructured the manuscript to place a primary emphasis on the novel Macromonas species that we have identified. These findings significantly enhance our comprehension of microbial diversity and the interactions between phages and hosts within freshwater ecosystems. Moreover, our results underscore the promising prospects for the discovery of new species and the advancement of phage biology studies in this specific context.
Reviewer 2 Report
The article “Macromonas nakdongensis sp. nov., isolated from freshwater and characterization of bacteriophage BK-30P, the first phage that infects genus” is devoted to description of new bacterial species. Due to the increasing number of works on the discovery of new microorganisms by assembling genome from the metagenome, it become important to obtain pure cultures of previously unknown bacterial species. Authors used current and classical methods of biology. The article can be published, but I have some comments.
1. As of 1 January 2001, the description of a new species must include the designation of a type strain (see Rule 18a), and a viable culture of that strain must be deposited in at least two publicly accessible collections in different countries and must be available. Evidence must be presented that the cultures are present, viable, and available at the time of publication. (International Code of Nomenclature of Prokaryotes Prokaryotic Code (2008 Revision https://doi.org/10.1099/ijsem.0.000778).
The registration numbers specified in the article (KCTC 52161 = JCM 31376 = FBCC-B1) were not found by us in the corresponding collections. It might be better described as a candidate species?
2. The foto here are in the article (Supplementary materials) are of poor quality.
3. M. bipunctata is a sulfur bacterium, the cell contains 4 types of inclusions (Bergey's 2005). It is unclear whether the isolated strain of the same genus possesses similar properties.
4. The authors named the strain Macromonas nakdongensis, but there is Macromonas aquatica in figure 1. The number KU360711 in NCBI database specifies Macromonas sp. BK-30 without a species name. Correct please.
5. The database for this genome indicated (https://www.ncbi.nlm.nih.gov/datasets/genome/GCF_002837145.1/):
Quality analysis
CheckM analysis (v1.2.2)
Completeness: 96.22% (100th Percentile)
1. Contamination: 1.99%
2What caused the contamination? Perhaps, the culture of strain is not pure?
Author Response
# Reviewer 2
The article “Macromonas nakdongensis sp. nov., isolated from freshwater and characterization of bacteriophage BK-30P, the first phage that infects genus” is devoted to description of new bacterial species. Due to the increasing number of works on the discovery of new microorganisms by assembling genome from the metagenome, it become important to obtain pure cultures of previously unknown bacterial species. Authors used current and classical methods of biology. The article can be published, but I have some comments.
- As of 1 January 2001, the description of a new species must include the designation of a type strain (see Rule 18a), and a viable culture of that strain must be deposited in at least two publicly accessible collections in different countries and must be available. Evidence must be presented that the cultures are present, viable, and available at the time of publication. (International Code of Nomenclature of Prokaryotes Prokaryotic Code (2008 Revision https://doi.org/10.1099/ijsem.0.000778).
The registration numbers specified in the article (KCTC 52161 = JCM 31376 = FBCC-B1) were not found by us in the corresponding collections. It might be better described as a candidate species?
→ (Response) We have deposited a new Macromonas strain into two culture collections. Since the thesis has not been published yet, conducting a search on the site is not currently feasible. We have enclosed deposit receipts from three culture collections.
- The foto here are in the article (Supplementary materials) are of poor quality.
→ (Response) We have made modifications to the images in the supplementary materials.
- M. bipunctata is a sulfur bacterium, the cell contains 4 types of inclusions (Bergey's 2005). It is unclear whether the isolated strain of the same genus possesses similar properties.
→ (Response) We tested the growth of strain BK-30T against four types of reduced and elemental sulfur. The results are written in the text. In addition, the presence of a Sox gene related to sulfur oxidation was confirmed through genome analysis of the BK-30T strain.
- The authors named the strain Macromonas nakdongensis, but there is Macromonas aquatica in figure 1. The number KU360711 in NCBI database specifies Macromonas sp. BK-30 without a species name. Correct please.
→ (Response) At the time of depositing in the NCBI database, the species name had not been finalized. Therefore, the new notation used was Macromonas sp. If the thesis is accepted, we intend to rectify the species name to Macromonas nakdongensis in the NCBI database. Furthermore, we will ensure alignment between the species name mentioned in the manuscript and supplementary materials, designating it as Macromonas nakdongensis.
- The database for this genome indicated (https://www.ncbi.nlm.nih.gov/datasets/genome/GCF_002837145.1/): Quality analysis CheckM analysis (v1.2.2) Completeness: 96.22% (100th Percentile). Contamination: 1.99%, What caused the contamination? Perhaps, the culture of strain is not pure?
→ (Response) Thank you for your insightful comments. We have conducted a comprehensive genome analysis of strain BK-30. The challenge arose from its slow growth rate and the production of small colonies, impeding the acquisition of sufficient biomass for thorough genome analysis. Consequently, we augmented the biomass by cultivating the cells in a broth medium, followed by cell acquisition through a 0.2μm filtration process. This approach resulted in a marginally lower genome completeness. However, it is crucial to emphasize that BK-30 has been deposited in two culture collections: KCTC in Korea and JCM in Japan, underscoring its verified pure culture status. While acknowledging your valid concerns, we wish to confidently affirm that BK-30 is indeed a pure culture
